# Factors of Non-Compliance with a Protocol for Oral Administration of Misoprostol (Angusta^®^) 25 Micrograms to Induce Labor: An Observational Study

**DOI:** 10.3390/jcm12041521

**Published:** 2023-02-14

**Authors:** Mathilde Pambet, Amélie Delabaere, Claire Figuier, Céline Lambert, Aurélie Comptour, Marion Rouzaire, Denis Gallot

**Affiliations:** 1Obstetrics and Gynaecology Department, CHU Clermont-Ferrand, 63000 Clermont-Ferrand, France; 2Biostatistics Unit, DRCI, CHU Clermont-Ferrand, 63000 Clermont-Ferrand, France; 3CIC 1405 CRECHE Unit, INSERM, Obstetrics and Gynaecology Department, CHU Clermont-Ferrand, 63000 Clermont-Ferrand, France; 4“Translational Approach to Epithelial Injury and Repair” Team, CNRS 6293, Inserm 1103, GReD, Auvergne University, 63000 Clermont-Ferrand, France

**Keywords:** Angusta^®^, misoprostol, oral, non-compliance, induction of labor (IOL)

## Abstract

We set out to identify factors of non-compliance with a protocol for the oral administration of misoprostol 25 µg (Angusta^®^) every 2 h (up to eight tablets), for the induction of labor (IOL). We conducted a retrospective study on IOL at term, on singleton pregnancies from 2019 to 2021, in a university hospital. The study included 195 patients, comprising 144 compliant protocols. Pain was statistically more frequent in the non-compliance group (92.2% vs. 62.5%, *p* < 0.001), and when a midwife was unavailable (15.7% vs. 0.7%, *p* < 0.001). A multivariable analysis found factors of good response (defined as going into labor before the administration of the median number of tablets, i.e., six) to be an indication for PROM (OR: 12.03, 95% CI: 5.42–26.71), and gestational age at induction (OR: 1.54, 95% CI: 1.19–2.01), independently of BMI, initial Bishop score, and parity. Patients with pain who were able to follow the protocol delivered 9 h earlier than patients with pain who interrupted the protocol and 16 h earlier than patients who experienced no pain. We identified two key elements that favored compliance: (i) providing the next tablet in advance; and (ii) offering patients early epidural analgesia when in pain in order to continue the protocol and go into labor promptly.

## 1. Introduction

According to the 2016 national perinatal survey [1], the rate of induced labor from 37 gestational weeks was 22% of deliveries in France. The methods for inducing labor with cervical ripening are placement of an intracervical catheter, vaginal prostaglandin E2 dinoprostone (Prostine^®^, vaginal gel and Propess^®^, vaginal diffusion system), intracervical prostaglandin (Prepidil intracervical^®^), and misoprostol. Among the cervical ripening techniques used during induction of labor (IOL), intravaginal dinoprostone was used in 90% of cases [1]. Misoprostol, a synthetic analogue of natural prostaglandin E1, has been used for IOL for almost 30 years. The action of this synthetic analogue of prostaglandin E1 is to produce cervical maturation and uterine contractions. Misoprostol may be administered vaginally, sublingually, or orally. There has been an increase in the use of oral preparations for IOL owing to its good acceptance and simplicity of use. Oral misoprostol is rapidly metabolized to an active metabolite, misoprostolic acid. Maximum plasma concentrations of misoprostolic acid are reached in 34 min [2]. Its mean plasma half-life is approximately 45 min through renal elimination. Misoprostol is being used more widely today for its efficacy, low cost, and stability at room temperature compared to various methods of administration [3].

Several studies comparing the efficacy of prostaglandins for cervical ripening suggest that misoprostol is more effective than prostaglandin E2 (PGE2), with a significant decrease in cesarean section rates [4,5].

A randomized controlled study by Shetty et al. in 2001 demonstrated that time between IOL and vaginal delivery was significantly shorter in a vaginal misoprostol group than in an oral misoprostol group. There was no difference in the mode of delivery, analgesic requirements, or neonatal outcomes between the two groups. However, there was a higher incidence of uterine hyperstimulation in the vaginal group and more cesarean sections were performed for fetal distress in this group [6]. A meta-analysis showed that low-dose (<50 μg) titrated oral misoprostol solution gave the lowest probability of cesarean section, whereas vaginal misoprostol (≥50 μg) gave the highest probability of achieving a vaginal delivery within 24 h [7]. Low-dose oral misoprostol solution (20 μg) administered every 2 h was at least as effective as both vaginal dinoprostone and vaginal misoprostol, with lower rates of cesarean delivery and uterine hyperstimulation [7]. A recent meta-analysis including 33 randomized clinical trials showed that oral misoprostol increased the duration of labor by 0.40 h, the risk of hypertonus, PROM, oxytocin need, and cesarean fever compared to vaginal misoprostol and decreased the risk of neonatal death, tachysystole, uterine hyperstimulation, preeclampsia, non-FHR, and abortion [8].

The advantages of the oral route over vaginal administration include ease of administration, better mobility, the ability to administer repeated doses of the drug without internal examinations, and the possible association with lower uterine hyperstimulation rates. Furthermore, misoprostol lowers the risk of sepsis in patients with premature rupture of the membranes (PROM) [9]. More than 80% of patients were satisfied with oral inductions, and 64% of patients stated that they would prefer to have the inducing agent given orally if they were to have another induction [10]. A French prospective cohort study of 520 women who chose their preferred method for labour induction showed that 67.5% of women chose oral misoprostol compared to 21% who chose a PGE2 pessary and 11.5% who chose a Foley catheter [11].

The tablet form presented no significant difference in the rate of patients delivering vaginally within 24 h compared to the sublingual form. Patients preferred the tablet form to the sublingual form, which they found had an unpleasant taste and consistency [12].

Angusta^®^ (oral misoprostol) is authorized in eight countries in Europe. The French marketing authorization was cleared by the ANSM (French national agency for the safety of health products, Saint-Denis, France) on 19 January 2018, and received final approval by the HAS (the French national authority for health, SaintDenis, France) on 18 April 2018. The recommended protocol for Angusta^®^ is 25 μg orally every 2 h or 50 μg orally every 4 h. The maximum dose is 200 μg over 24 h. The optimal dose allows a short induction-to-birth interval and effective uterine contractions preserving fetal wellbeing without increasing the rate of uterine hyperstimulation. A Cochrane systematic review suggested that the optimal dosage of oral misoprostol was 20–25 µg given every 2 h [13]. After eight tablets, efficacy was no higher and there was a risk of adverse effects [14].

We evaluated the factors of non-compliance with the protocol of oral misoprostol administration in tablet form (Angusta^®^) in a French tertiary delivery unit. We also investigated the clinical factors influencing the response to Angusta^®^ and the impact of the protocol on the delivery route, and we analyzed maternal and fetal variables.

## 2. Materials and Methods

We conducted a retrospective observational study on IOL at term, using Angusta^®^, at the University Hospital of Clermont-Ferrand, France, from March 2019 to March 2021.

### 2.1. Participants

Inclusion criteria were IOL using Angusta^®^ at physicians’ discretion for singleton pregnancies, live fetus in cephalic presentation, a Bishop score ≤ 6 and a gestational age ≥ 37 weeks. Exclusion criteria were in utero fetal death, breech presentation, positive history of uterine surgery including cesarean section, allergy or hypersensitivity to the active substance, severe growth restriction (lower than third percentile with Doppler anomalies), uterine malformation, low-lying or previa placenta, unexplained bleeding, and renal insufficiency (glomerular filtration rate < 15 mL/min/1.73 m^2^).

Indications for IOL with misoprostol were prolonged pregnancy (gestational age ≥ 41–41 + 6, weeks + days), PROM, diabetes, suspicion of fetal macrosomia, gestational hypertension and preeclampsia, oligohydramnios, antithrombotic therapeutic window, and other indications such as psychological distress.

### 2.2. Study Procedures

The Angusta^®^ protocol consisted of the oral administration of 25 µg of misoprostol every 2 h. The dose not to be exceeded in 24 h was 200 µg, or eight tablets. The entire procedure took 14 h. All of the tablets were delivered with a glass of water by a midwife. Tablet administration was recorded on a handwritten sheet. Pain was scored on a numerical scale at tablet administration. Cardiotocography (CTG) was performed for at least 30 min before and 60 min after first the administration of misoprostol. In the absence of an anomaly, repetitive CTG was not systematic. CTG was performed only in the event of painful uterine contractions suggesting labor, loss of amniotic fluid, or bleeding.

There was an additive effect of misoprostol and oxytocin. This required a wash-out interval of 4 h between the last intake of Angusta^®^ and the introduction of oxytocin.

We recorded each patient’s age, gestity, parity, body mass index (BMI), initial Bishop score, and gestational age of IOL.

The beginning of labor required cervical dilation greater than 3 cm, and active labor was diagnosed as a cervical dilation greater than 6 cm. We reported the time between taking the first tablet and the start of labor, and between taking the first tablet and the active phase of labor, in hours.

### 2.3. Data Collected

For all patients, we recorded the number of tablets taken and calculated the degree of compliance with the protocol as a percentage every 2 h (up to 3 h allowed). The administration was recorded as compliant if 100% of the tablets were administered, and as delayed if fewer than 100% of the tablets were administered. For example, if a patient needed six tablets to go into labor, but only four tablets were taken, we had 66.7% protocol compliance. At each interruption of the protocol, we noted whether this was due to pain, PROM, or a midwife being time-constrained. If a tablet was missed owing to pain, we recorded the Bishop score, the numerical pain scale score, the numbers of tablets taken, and whether the patient needed to start a treatment such as nalbuphine.

Epidural analgesia was available on request to all patients in labor. Mode of delivery (spontaneous, instrumental, or cesarean) was recorded, together with the time between the first tablet and delivery.

Selected post-labor outcome variables for the mother were blood loss, post-partum hemorrhage, and indication for cesarean section. The outcome variables for the neonates were median Apgar score at 5 min, percentage Apgar score < 7 at 5 min, mean umbilical artery pH, percentage of umbilical artery pH < 7.15, weight at birth, and transfer to an intensive care unit.

A good response to the protocol was defined as going into labor before the administration of the median number of tablets (six).

### 2.4. Statistical Analysis

The statistical analysis was performed using Stata software (version 15; StataCorp, College Station, TX, USA). All tests were two-sided, with an alpha level set at 5%. Categorical variables were expressed as the number of patients and associated percentages, and continuous variables as mean ± standard deviation or median [25th; 75th percentiles], according to the statistical distribution. The rate of compliant administration was presented with a 95% confidence interval (95% CI). Comparisons between independent groups (nulliparous vs. multiparous, compliant vs. delayed administration, and good vs. delayed response) were made using the chi-squared test or Fisher’s exact test for categorical variables, and a Student’s *t* test or the Mann-Whitney *U* test for continuous variables. A multivariable logistic regression was performed to study factors associated with good response (<6 tablets), selecting explanatory variables according to univariate results and clinical relevance: gestational age at induction, BMI, initial Bishop score, indication for PROM, and parity (nulliparous vs. multiparous). The results were expressed as odds ratio (OR) and 95% CI. Finally, the sample was divided into four groups (“no pain and compliant administration” vs. “no pain and delayed administration” vs. “pain and compliant administration” vs. “pain and delayed administration”), which were subsequently compared by the Fisher’s exact test for categorical variables and by the Kruskal-Wallis test for continuous variables. If appropriate (omnibus *p*-value less than 0.05), a *post hoc* multiple comparisons test was performed (Marascuilo for categorical variables or Dunn’s for continuous variables).

## 3. Results

In all, 195 consecutive patients were recruited for the study over two years. Of these, 141 were nulliparous and 54 were multiparous (Figure 1).

Nulliparous patients were significantly younger than multiparous patients (28.9 ± 5.4 years vs. 33.8 ± 4.8 years, *p* < 0.001). There was no difference between the nulliparous and multiparous patients in terms of gestational age, BMI, or initial Bishop score at the beginning of induction. Regarding the indications for IOL, nulliparous patients were induced in 43.3% of cases for PROM, in 19.1% of cases for post-term, and in 19.1% of cases for gestational diabetes. In the multiparous group, 29.6% of the patients were induced for gestational diabetes or fetal macrosomia, and 25.9% were induced for PROM (Table 1).

Of the 195 patients included, 144 had compliant administration of the tablets (73.8%, 95% CI: 67.1–79.9%). Patient’s age, BMI or parity were not significantly associated with compliant administration, and neither were the initial Bishop score, gestational age at induction, or the number of tablets taken (Table 2). Pain was statistically more frequent in the group with delayed administration than in the group with compliant administration (92.2% and 62.5% respectively, *p* < 0.001), and more intense (7.35 ± 1.63 vs. 5.67 ± 2.96 respectively, *p* < 0.001). In 4.6% of cases, a midwife being time-constrained was reported, leading to delayed administration in eight cases and compliant administration in one case (tablet administration before 3 h) (*p* < 0.001). The occurrence of a rupture of membranes (ROM) during the protocol was not associated with the compliant administration (*p* = 0.36). The use of nalbuphine was significantly more frequent in the delayed administration group.

There was no significant difference for going into labor before five tablets between patients who had compliant administration and those with delayed administration (*p* = 0.34). The time between taking the first misoprostol tablet and the beginning of labor was significantly shorter in the group with compliant administration (14 h [9; 25] vs. 19 h [13; 30], *p* = 0.004). The results were similar for the time between taking the first misoprostol tablet and the active labor phase (17 h [11; 28] vs. 23 h [17; 30], *p* = 0.005). The time from IOL to delivery differed between the two groups and was shorter in the group with compliant administration (*p* = 0.004) (Table 2).

A good response to the protocol was achieved in 86 patients (44.1%), who went into labor before the administration of six tablets, i.e., the median number of tablets. Age, BMI, parity, gestational age at induction, and the occurrence of ROM during the treatment were not associated with the response to this protocol. The indication for IOL emerged as a factor influencing the protocol response (*p* < 0.001). In particular, the indication for PROM was higher in the group with good response (68.2% vs. 19.3%). Gestational age was higher in patients with PROM who took fewer than six tablets (*p* = 0.009). Pain during the protocol and pain intensity were significantly associated with good response to the protocol. Having a more advanced Bishop score at the IOL was found to be a factor indicating good response to the protocol (*p* = 0.03) (Table 3).

A multivariable analysis was performed by taking variables known at the beginning of IOL to identify the factors of good response to this protocol. The variables were as follows: gestational age at induction, BMI, initial Bishop score, indication for PROM, and parity. The indication for PROM was significantly associated with a good response to this protocol (OR: 12.03, 95% CI: 5.42–26.71, *p* < 0.001), as was the gestational age at induction (OR: 1.54, 95% CI: 1.19–2.01, *p* = 0.001). However, being nulliparous emerged as a factor of poor response to the protocol (OR: 0.45, 95% CI: 0.21–0.99, *p* = 0.05) (Table 4).

An additional analysis was performed comparing four groups of patients, defined according to the presence/absence of pain and the compliant/delayed administration of the tablets. First, in the group with no pain and delayed administration, two patients were non-compliant owing to the midwife being time-constrained. In the group with pain and compliant administration, the times between the first tablet and the beginning of labor, the active phase of labor, and delivery were significantly shorter than in the group with pain and delayed administration (*p* < 0.001). Furthermore, having no pain and compliant administration vs. having pain and delayed administration lengthened the time of IOL until childbirth by about 16 h. Complying with the protocol while in pain was a factor of good response (Table 5).

In this cohort, there was 79.0% vaginal birth and 21.0% cesarean section, and 94.9% of patients had analgesia by epidural. There was no relation between compliance with the protocol and delivery route (*p* = 0.61). We looked at the different indications for cesarean section, whether due to a failure of induction (cervical dilation less than 3 cm, 24.4%), stagnation at 3 cm (7.3%), and whether at the beginning of labor (cervical dilation 3–6 cm, 26.8%), or during the active phase of labor (cervical dilation greater than or equal to 6 cm, 41.5%). In the group with compliant administration, there were as many cesarean sections for failure of induction as during the active phase of labor (34.5%). However, in the group with delayed administration, there was no cesarean for failure of induction, and more than half of the cesareans were during the active phase of labor (58.3%) (Table 6).

The maternal and neonatal variables did not differ between compliant and delayed administration groups (Table 6).

## 4. Discussion

This observational study found evidence that compliance with a protocol for IOL in a tertiary maternity unit was possible. Pain was the first non-compliance factor for our protocol. We know that misoprostol is used to induce uterine contractions, and that in many studies it is even responsible for uterine hyperstimulation when taken vaginally compared to orally [15]. In our protocol, we conducted CTG if our patients were in pain, but because of a lack of data we could not find any occurrence of hyperstimulation. This could be an outcome of interest to evaluate in future studies. We also resorted to analgesia (nalbuphine) in 36% of cases. These two variables were statistically more frequent in the group with delayed administration, because this group was more often in pain. The literature reports that in an oral misoprostol group, 16% needed tocolysis, and only 2% needed opioid-like nalbuphine vs. 32% with misoprostol vaginal insert [16]. It is difficult to compare this result to ours, because we used a different oral misoprostol protocol, but it argues for better pain assessment.

The delayed administration of tablets by midwives also proved to be a factor of non-compliance with our protocol. It led to the protocol being modified during our study, when patients were progressively empowered to take their own tablets. Pain assessment and medication schedules were filled in by the patients themselves. For example, in Denmark, IOL has been performed in low-risk pregnancies at home since 2016, and delivery is achieved within 48 h for 70.1% of women [17]. It might one day be possible to envisage outpatient IOL to increase maternal satisfaction and comfort and reduce length of antenatal hospital stay [18]. We recommend that the midwife delivers the tablet to be taken immediately, together with one tablet provided in advance for the next dose in order to anticipate time constraints.

Median time to delivery was significantly shorter in the group with compliant administration (23 h vs. 29 h, *p* = 0.004). In the literature, there is still a lack of data to establish a clear protocol with a precise dose and frequency of administration [19,20]. It is a strength of this study that all inductions were handled according to a predefined protocol and that a misoprostol tablet manufactured for IOL with a known content of active agent was used. It would be useful to standardize the protocols, although Weeks et al., in an article summarizing the different misoprostol-based protocols, concludes that “Ideally, formal pharmacokinetic studies would help to clarify the differences between tablets and oral solution, and to establish the optimal frequency of the lowest effective dose. Regrettably, such studies are unlikely to be supported by pharmaceutical companies as misoprostol is too cheap to justify the investment” [21].

In our first analysis, we found that patients with a higher initial Bishop score had a better response to the protocol, but after our multivariable analysis we did not find this variable to be a factor of good response, which seems surprising because every IOL is more effective on a ripe cervix [22].

Our secondary analysis showed that patients induced for PROM had a better response to the protocol, and even more so if their term at PROM was advanced. Only a few trials have reported data with PROM for IOL with oral misoprostol: “Membrane status brought an additional significant impact on time-to-vaginal delivery as main outcome measure of efficacy” [23]. Our cohort only included 75 patients with PROM, and thus the power was insufficient to draw certain conclusions. However, it could be of interest to select the indication for induction to appraise its efficacy, especially since no one method for induction has yet demonstrated its superiority.

BMI was not found to be a factor of good response to the protocol. The impact of obesity on IOL has been described by both Norman et al. and Lassiter, the latter who stated that “Obese women demonstrating a longer duration of labor for women with body mass index over 30 kg/m^2^, have a longer duration of induction to delivery, require more oxytocin to augment labor, and more doses of misoprostol” [24,25]. Our sample was probably too small to find a significant difference according to BMI.

We know that parity affects time to delivery: “The percentage of vaginal delivery within 24 h is higher in multiparous women than in nulliparous” [23]. As in the literature, with our multivariable analysis, nulliparity seemed to be a factor indicating poor response to the protocol.

The occurrence of pain and nalbuphine treatment were factors of good response to this protocol, which means the probable start of labor for these patients. The advantages of nalbuphine include analgesia with smaller drug doses, resulting in a lower incidence of side effects at the first stage of labor, without reducing the number of contractions [26,27].

We demonstrated that pain was a good response factor. Being in pain proves the efficacy of the protocol with uterine contractions, and the probable start of labor. However, pain also emerged as a factor of non-compliance with the protocol. We saw in our study that being in pain but with compliant administration reduced the time until the active phase of labor and childbirth compared to patients with pain and delayed administration. Labor was shortened by 10 h if the administration was compliant despite pain. It is therefore important to continue the protocol even when in pain. We advocate earlier epidural analgesia to continue the protocol with pain relief and thereby increase the chances of going into labor promptly.

We had 79% vaginal deliveries and 21% cesarean sections. There was no statistical correlation between protocol compliance and delivery route. Our data can be superimposed on the literature, allowing for the fact that three-quarters of our cohort were nulliparous. Cesarean sections were not more numerous with titrated oral misoprostol compared with all the other different ways to induce labor [28,29,30]. Maternal and neonatal outcomes were similar in our study. The oral administration of misoprostol has already been the subject of several studies, proving its efficacy and its maternal and neonatal safety [31,32].

This study, as with all observational retrospective studies, has limits inherent to its design, including the lack of a double blind. Furthermore, our sample was small and single-center. The strength of the study is that it focuses on one clear protocol, with a manufactured tablet, to find how to ameliorate compliance in a tertiary university hospital. The most important factor inducing non-compliance was pain, and therefore it will be necessary to add pain management to our protocol.

## 5. Conclusions

The oral administration of misoprostol 25 µg, every 2 h, up to eight tablets, for the IOL was complied with in more than 73% of cases in a tertiary maternity unit, without affecting maternal or neonatal outcomes. Among the recorded data including gestational age, BMI, initial bishop score, parity, and the indication for IOL, the factors of non-compliance with the protocol were pain and delayed administration of the tablets. We showed that a patient with pain who still followed the protocol had a greater likelihood of going into labor, and into the active phase of labor, and of giving birth more quickly. We identified two key elements favoring compliance: (i) providing the next tablet in advance (to anticipate the midwife being time-constrained, an occurrence responsible for 5% of delayed protocol), and (ii) offering patients early epidural analgesia when they are in pain, and so continuing the protocol to go into labor promptly. Oral misoprostol appears more effective for patients with PROM after 39 weeks.

## Figures and Tables

**Figure 1 jcm-12-01521-f001:**
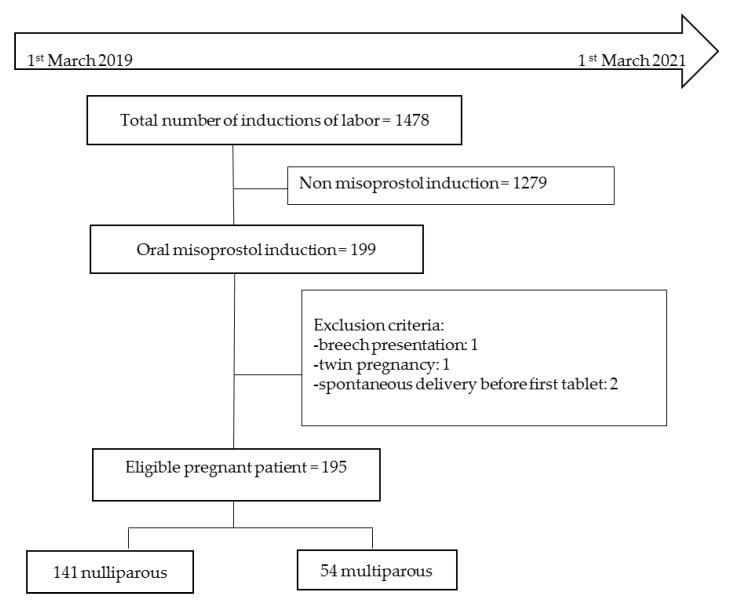
Flow chart.

**Table 1 jcm-12-01521-t001:** Demographic and clinical maternal baseline data.

	Total (*n* = 195)	Nulliparous (*n* = 141)	Multiparous (*n* = 54)	*p*
Age (years)	30.3 ± 5.7	28.9 ± 5.4	33.8 ± 4.8	<0.001
GA at induction (weeks)	39.6 ± 1.4	39.7 ± 1.4	39.3 ± 1.4	0.08
Body mass index (kg/m^2^)	25.9 ± 6.0	25.7 ± 5.8	26.4 ± 6.5	0.45
Initial Bishop score (*n* = 193)	3 [2; 4]	3 [2; 4]	3 [2; 3]	0.89
Indication for IOL				0.003
Post-term (≥41 WG ± 6)	36 (18.5)	27 (19.1)	9 (16.7)	
GD/fetal macrosomia	43 (22.1)	27 (19.1)	16 (29.6)	
Hypertension/preeclampsia	11 (5.6)	11 (7.8)	0 (0.0)	
Antithrombotic therapy	9 (4.6)	4 (2.8)	5 (9.3)	
PROM	75 (38.5)	61 (43.3)	14 (25.9)	
Others ^a^	21 (10.8)	11 (7.8)	10 (18.5)	

Data are presented as number of patients (percentages), mean ± standard deviation, or median [25th; 75th percentiles]. GA: gestational age; GD: gestational diabetes; IOL: induction of labor; PROM: premature rupture of membrane; WG: weeks of gestation. ^a^: IOL for convenience of psychiatric disorder.

**Table 2 jcm-12-01521-t002:** Factors associated with protocol compliance.

	Total (*n* = 195)	Compliant Administration (*n* = 144)	Delayed Administration (*n* = 51)	*p*
Age (years)	30.3 ± 5.7	30.4 ± 5.7	29.9 ± 5.7	0.63
GA at induction (weeks)	39.6 ± 1.4	39.6 ± 1.4	39.7 ± 1.3	0.64
Body mass index (kg/m^2^)	25.9 ± 6.0	26.1 ± 6.1	25.4 ± 5.7	0.47
Initial Bishop score (*n* = 193)	3 [2; 4]	3 [2; 4]	3 [1; 3]	0.62
Indication for IOL				0.72
Post-term (≥41 WG ± 6)	36 (18.5)	27 (18.7)	9 (17.7)	
GD/fetal macrosomia	43 (22.0)	35 (24.3)	8 (15.7)	
Hypertension/preeclampsia	11 (5.6)	7 (4.9)	4 (7.8)	
Antithrombotic therapy	9 (4.6)	7 (4.9)	2 (3.9)	
PROM	75 (38.5)	52 (36.1)	23 (45.1)	
Others ^a^	21 (10.8)	16 (11.1)	5 (9.8)	
Nulliparous	141 (72.3)	104 (72.2)	37 (72.5)	0.96
Number of tablets	6 [4; 8]	6 [4; 8]	6 [4; 8]	0.55
Occurrence of pain	137 (70.3)	90 (62.5)	47 (92.2)	<0.001
Pain VAS (/10)	6.10 ± 2.78	5.67 ± 2.96	7.35 ± 1.63	<0.001
ROM during protocol	14 (7.2)	12 (8.3)	2 (3.9)	0.36
Unavailable midwife	9 (4.6)	1 (0.7)	8 (15.7)	<0.001
Nalbuphine injection	70 (35.9)	43 (29.9)	27 (52.9)	0.003
Salbutamol injection	51 (26.2)	23 (16.0)	28 (54.9)	<0.001
Labor before five tablets	60 (30.8)	47 (32.6)	13 (25.5)	0.34
Time first tablet to 3 cm cervical dilation (hours) (*n* = 181)	16 [10; 26]	14 [9; 25]	19 [13; 30]	0.004
Time first tablet to 6 cm cervical dilation (hours) (*n* = 171)	18 [13; 29]	17 [11; 28]	23 [17; 30]	0.005
Time first tablet to delivery (hours)	26 [18; 34]	23 [16; 34]	29 [23; 36]	0.004

Data are presented as number of patients (percentages), mean ± standard deviation, or median [25th; 75th percentiles]. GA: gestational age; GD: gestational diabetes; IOL: induction of labor; PROM: premature rupture of membrane; ROM: rupture of membranes; VAS: visual analogue scale, WG: weeks of gestation. ^a^: IOL for convenience or psychiatric disorder.

**Table 3 jcm-12-01521-t003:** Factors associated with the response to the protocol.

	Good Response ^a^ (*n* = 86)	Delayed Response ^b^ (*n* = 109)	*p*
Age (years)	30.4 ± 5.2	30.1 ± 6.1	0.66
GA at PROM (weeks) (*n* = 75)	39.4 ± 1.2	38.1 ± 1.9	0.009
GA at induction (weeks)	39.8 ± 1.3	39.5 ± 1.4	0.11
Body mass index (kg/m^2^)	25.2 ± 5.7	26.5 ± 6.2	0.13
Initial Bishop score (*n* = 193)	3 [2; 4]	2 [1; 3]	0.03
Indication for IOL			<0.001
Post-term (≥41 WG ± 6)	12 (14.0)	24 (22.0)	
GD/fetal macrosomia	8 (9.3)	35 (32.1)	
Hypertension/preeclampsia	1 (1.2)	10 (9.2)	
Antithrombotic therapy	5 (5.8)	4 (3.7)	
PROM	54 (62.8)	21 (19.3)	
Others ^c^	6 (7.0)	15 (13.8)	
Nulliparous	61 (70.9)	80 (73.4)	0.70
Occurrence of pain	85 (98.8)	52 (47.7)	<0.001
Pain VAS (/10)	7.41 ± 1.49	5.08 ± 3.11	<0.001
ROM during protocol	6 (7.0)	8 (7.3)	1.00
Unavailable midwife	3 (3.5)	6 (5.5)	0.73
Nalbuphine injection	42 (48.8)	28 (25.7)	0.001
Salbutamol injection	23 (26.7)	28 (25.7)	0.87
Epidural	81 (94.2)	104 (95.4)	0.75
Vaginal birth	71 (82.6)	83 (76.1)	0.28

Data are presented as number of patients (percentages), mean ± standard deviation, or median [25th; 75th percentiles]. GA: gestational age; GD: gestational diabetes; IOL: induction of labor; PROM: premature rupture of membrane; ROM: rupture of membranes; VAS: visual analogue scale; WG: weeks of gestation. ^a^: patients who went into labor before the administration of six tablets; ^b^: patients who required more than six tablets; ^c^: IOL for convenience or psychiatric disorder.

**Table 4 jcm-12-01521-t004:** Multivariable analysis of factors associated with good response (taking fewer than six tablets before labor) to the protocol.

	OR	95% CI	*p*
GA at induction	1.54	1.19–2.01	0.001
Body mass index	0.98	0.92–1.03	0.45
Initial Bishop score	1.04	0.82–1.31	0.76
Indication for PROM	12.03	5.42–26.71	<0.001
Nulliparous	0.45	0.21–0.99	0.05

CI: confidence interval; GA: gestational age; PROM: premature rupture of membrane; OR: odds ratio.

**Table 5 jcm-12-01521-t005:** Labor progression timeline according to pain and protocol compliance.

	No Pain		Pain		
	Compliant Administration (*n* = 36)	Delayed Administration (*n* = 2)	Compliant Administration (*n* = 108)	Delayed Administration (*n* = 49)	*p*
Number of tablets < 6	1 (2.8)	0 (0.0)	63 (58.3)	22 (44.9)	<0.001 ^abcd^
Time first tablet to 3 cm cervical dilation (hours) (*n* = 181)	28 [23; 31]	32 [32; 32]	12 [8; 17]	19 [13; 27]	<0.001 ^abce^
Time first tablet to 6 cm cervical dilation (hours) (*n* = 171)	32 [27; 37]	35 [35; 35]	14 [10; 20]	22 [16; 28]	<0.001 ^abce^
Time first tablet to delivery (hours)	35 [30; 39]	42 [41; 43]	19 [15; 28]	29 [23; 34]	<0.001 ^abce^

Data are presented as number of patients (percentages) or median [25th; 75th percentiles]. ^a^: significant difference (*p* < 0.05) between “Pain and compliant administration” and “No pain and compliant administration”; ^b^: significant difference (*p* < 0.05) between “Pain and delayed administration” and “No pain and compliant administration”; ^c^: significant difference (*p* < 0.05) between “Pain and compliant administration” and “No pain and delayed administration”; ^d^: significant difference (*p* < 0.05) between “Pain and delayed administration” and “No pain and delayed administration”; ^e^: significant difference (*p* < 0.05) between “Pain and delayed administration” and “Pain and compliant administration”.

**Table 6 jcm-12-01521-t006:** Relation between compliance with protocol and delivery route, and postpartum and neonatal outcomes.

	Total (*n* = 195)	Compliant Administration (*n* = 144)	Delayed Administration (*n* = 51)	*p*
Delivery route				
Vaginal birth	154 (79.0)	115 (79.9)	39 (76.5)	0.61
Instrumental delivery	39 (20.0)	28 (19.4)	11 (21.6)	0.75
Cervical dilation ^a^				0.046
<3 cm	10/41 (24.4)	10/29 (34.5)	0/12 (0.0)	
=3 cm	3/41 (7.3)	1/29 (3.4)	2/12 (16.7)	
>3 cm to <6 cm	11/41 (26.8)	8/29 (27.6)	3/12 (25.0)	
≥6 cm	17/41 (41.5)	10/29 (34.5)	7/12 (58.3)	
Epidural	185 (94.9)	134 (93.1)	51 (100.0)	0.07
Blood loss (mL)	170 [100; 420]	150 [100; 410]	200 [100; 450]	0.50
Blood loss ≥ 500 mL	39 (20.0)	28 (19.4)	11 (21.6)	0.75
Instrumental delivery	39 (20.0)	28 (19.4)	11 (21.6)	0.75
Umbilical artery pH	7.22 ± 0.09	7.22 ± 0.09	7.24 ± 0.07	0.11
Umbilical artery pH < 7.15	34 (17.8)	28 (20.0)	6 (11.8)	0.19
Apgar score at 5 min	10 [10; 10]	10 [10; 10]	10 [9; 10]	0.26
Apgar score < 7 at 5 min	6 (3.1)	5 (3.5)	1 (2.0)	1.00
Fetal weight (g)	3415 ± 408	3436 ± 427	3357 ± 344	0.19
Transfer to ICU	8 (4.1)	5 (3.5)	3 (5.9)	0.43

Data are presented as number of patients (percentages), mean ± standard deviation, or median [25th; 75th percentiles]. ICU: intensive care unit. ^a^ At cesarean indication.

## Data Availability

Not applicable.

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
