# Peer review of "Factors of Non-Compliance with a Protocol for Oral Administration of Misoprostol (Angusta®) 25 Micrograms to Induce Labor: An Observational Study"

_jcm, 2023, doi:10.3390/jcm12041521_

Round 1
Reviewer 1 Report
The benefits of low-dose misoprostol for labor induction are well established. The authors analyzed the effectiveness of the French manufacture tabbletes Angusta. The merit of the author, despite the small sample, is a scrupulous search for the causes of non-compliance and possible ways to overcome them.
The article is intended for practicing physicians. In conclusion, along with a description of the pill regimen, it would be nice to list the described factors ( including parity, BMI, timely pill intake, early analgesia due to unproductive pain, etc.). Taking into account them it is expected to improve favorable outcomes of IOL
Author Response
Thank you for your comments. We modified the conclusion part according to your comment.
Reviewer 2 Report
The authors conducted a retrospective study on factors of non compliance with a protocol of oral administration of misoprostol. The manuscript is well written and technically sound. The content is relevant and of interest to the reader, I therefor recommend to accept the manuscript. The manuscript is clearly structured and easy to read, the conclusions are consistent with the detailed results and arguments. The main question is address by the results and the authors contribute to the field with new facts.
Author Response
Thank you for your comments. English language has been reedited by a professional english medical writer.
Reviewer 3 Report
Interesting paper, minor language revisions are needed. Meanwhile, more recent references might be useful
Author Response
English language has been reedited by a professional english medical writer. Reference list has been updated (3 additional references published since 2022)